# The Underlying Role of the Glymphatic System and Meningeal Lymphatic Vessels in Cerebral Small Vessel Disease

**DOI:** 10.3390/biom12060748

**Published:** 2022-05-25

**Authors:** Yu Tian, Mengxi Zhao, Yiyi Chen, Mo Yang, Yilong Wang

**Affiliations:** 1Department of Neurology, Beijing Tiantan Hospital, Capital Medical University, Beijing 100070, China; rainietien@mail.ccmu.edu.cn (Y.T.); mxrdf0909@163.com (M.Z.); cyeeyeee@163.com (Y.C.); emmayang11111@outlook.com (M.Y.); 2Chinese Institute for Brain Research, Beijing 102206, China; 3China National Clinical Research Center for Neurological Diseases, Beijing 100070, China; 4Advanced Innovation Center for Human Brain Protection, Capital Medical University, Beijing 100070, China; 5National Center for Neurological Diseases, Beijing 100070, China

**Keywords:** cerebral small vessel disease, glymphatic system, meningeal lymphatic vessel, cerebrospinal fluid

## Abstract

There is a growing prevalence of vascular cognitive impairment (VCI) worldwide, and most research has suggested that cerebral small vessel disease (CSVD) is the main contributor to VCI. Several potential physiopathologic mechanisms have been proven to be involved in the process of CSVD, such as blood-brain barrier damage, small vessels stiffening, venous collagenosis, cerebral blood flow reduction, white matter rarefaction, chronic ischaemia, neuroinflammation, myelin damage, and subsequent neurodegeneration. However, there still is a limited overall understanding of the sequence and the relative importance of these mechanisms. The glymphatic system (GS) and meningeal lymphatic vessels (mLVs) are the analogs of the lymphatic system in the central nervous system (CNS). As such, these systems play critical roles in regulating cerebrospinal fluid (CSF) and interstitial fluid (ISF) transport, waste clearance, and, potentially, neuroinflammation. Accumulating evidence has suggested that the glymphatic and meningeal lymphatic vessels played vital roles in animal models of CSVD and patients with CSVD. Given the complexity of CSVD, it was significant to understand the underlying interaction between glymphatic and meningeal lymphatic transport with CSVD. Here, we provide a novel framework based on new advances in main four aspects, including vascular risk factors, potential mechanisms, clinical subtypes, and cognition, which aims to explain how the glymphatic system and meningeal lymphatic vessels contribute to the progression of CSVD and proposes a comprehensive insight into the novel therapeutic strategy of CSVD.

## 1. Introduction

Cerebral small vessel disease (CSVD) describes a set of syndromes that occurs when the perforating cerebral arterioles, capillaries, and venules are damaged, with the under-lying pathophysiology incompletely understood. It commonly causes up to 25% of strokes [1] and 45% of cognitive impairment or dementia [2], as well as physical and neuro-psychological disorders. The prevalence of CSVD increases with aging, affecting 5% of middle-aged and senior people and almost everyone over 90 years of age [3]. CSVD arises from different neuroimaging features on magnetic resonance imaging (MRI), including white matter hyperintensities (WMHs), enlarged perivascular spaces (EPVSs), cerebral microbleeds (CMBs), and brain atrophy [4]. CSVD has been regarded as a “dynamic whole-brain disease” with heterogeneous etiologies, diverse cerebrovascular mechanisms, and multifactorial pathologies [5]. Given the complexity and heterogeneity of CSVD, there is an urgent need for a novel overall understanding.

Recently, the glymphatic system (GS) and meningeal lymphatic vessels (mLVs) have been discovered to be solute and have fluid clearance systems in the central nervous system (CNS) [6,7,8]. With the significant functions in maintaining homeostasis of CNS, there was perhaps no doubt that the glymphatic-meningeal lymphatic system played a critical role in the pathophysiology of multiple central neurological system diseases, such as Alzheimer’s Disease (AD), Parkinson’s disease (PD), stroke, glioma, etc. Since the importance of glymphatic and lymphatic drainage in the brain has been thoroughly discussed in the context of neurodegenerative disorders and neuro-oncology [9], to our knowledge, few reviews focus on the role of glymphatic-meningeal lymphatic drainage in CSVD.

In fact, due to the different anatomy of the cerebral vessels and brain parenchyma between humans and rodents, there are still no perfect disease models that replicate only a part of the pathologies and pathophysiologies of CSVD. Most preclinical studies about the glymphatic system and meningeal lymphatic system in CSVD were based on rodent models examining aging, hypertension, diabetes, and internal carotid artery stenosis or occlusion, which provided limited information about the potential linking of the glymphatic-mLVs system to CSVD. Kounda et al. revealed that aging spontaneously hypertensive stroke prone rats (SHRSP), a most frequently used model for simulating the pathology of CSVD in human [10], showed significantly reduced cerebrospinal fluid (CSF) and interstitial fluid (ISF) transport and moderately decreased glymphatic function compared with Wistar-Kyoto rats [11]. A recent study illustrated the relationship between glymphatic function reflected by diffusion tensor image analysis along the perivascular space (DTI-ALPS) and neuroimaging markers, including WMH, lacunas, CMBs, and EPVS in basal ganglia, providing more convincing evidence about the glymphatic-meningeal lymphatic system in small vessel disease [12].

This review firstly describes a brief perspective of the glymphatic system, meningeal lymphatic system, CSF circulation, and their associations. Then, we discuss the current evidence on GS and mLVs involved in CSVD from four main aspects, including vascular risk factors, potential mechanisms, clinical subtypes, and dementia, to provide an overview linking glymphatic-mLVs system to small vessel disease. A better understanding of the underlying relevance of glymphatic and meningeal lymphatic system and CSVD, and potential pathologies and pathophysiologies between them, is needed to develop novel early phase markers and identify targets for disease treatments.

## 2. An Overall Perspective of the Glymphatic System and Meningeal Lymphatic Vessels

### 2.1. The Cerebrospinal Fluid Circulation

The cerebrospinal fluid, which is essential to maintain intracranial pressure (ICP) and metabolites clearance, displays a fundamental mechanical and metabolic role in the brain. Nowadays, CSF drainage remains a topic of debate. Surrounding most of the brain and spinal cord, CSF originates primarily from the choroid plexus within ventricles and mainly circulates from ventricular compartments to the subarachnoid space (SAS) [13]. Classically, CSF homeostasis was presumed to be achieved by “the traditional venous CSF outflow pathway”, in which CSF was thought to drain into the cerebral venous sinuses via arachnoid granulations [13]. The outflow of CSF across arachnoid granulation depended on the pressure gradient between SAS and the venous blood of the dural sinuses (Figure 1). The drainage system was engaged in 50–70 mm of CSF pressure and the flow rate was approximately 0.35 mL/min under a 3– to 4–mmHg pressure gradient in normal conditions. The estimated maximal draining velocity of this system was 1.0 mL/min [14].

In addition to the traditional view of reabsorption into venous sinuses by arachnoid granulation, alternative exit routes, namely cranial and spinal nerve sheath outflow pathways, have been very strongly verified in rodents and humans [15]. The mainly perineural drainage pathway was the olfactory outflow pathway from the cribriform plate to lymphatic vessels of the nasal mucosa into the nasal cavity [16]. Other CSF outflow routes along the cranial nerves included optic, trigeminal, facial, acoustic, glossopharyngeal, vagus, and accessory nerves [15,17,18,19]. The outflow of CSF also occurred from the spinal cord along spinal nerves. Thus, a consensus has emerged that there are CSF outflow pathways along cranial and spinal nerves, backed by a great deal of evidence. Lastly, but equally important, was lymphatic outflow pathways depending on the glymphatic system in the perivascular space and meningeal lymphatic vessels in the dura. This route was also a significant CSF efflux pathway, as described in the next section.

### 2.2. Virchow-Robin Spaces and the Glymphatic System

The earliest descriptions of Virchow-Robin spaces (VRSs), also known as PVSs, arose as early as the mid-1800s, when investigators, including Rudolf Virchow and Charles Robin, observed spaces surrounding blood vessels penetrating the brain parenchyma, including periarteriolar, pericapillary, and perivenular spaces [20,21]. Anatomically, VRSs presented region-specific diversity. The PVSs in basal ganglia represented the space between two layers of the leptomeninges, whereas for venules or arterioles entering brain parenchyma via convexity cortex, PVSs referred to the space sandwiched between one layer of leptomeningeal membrane and the brain itself [22]. Due to the anatomical heterogeneity, PVSs within basal ganglia were believed to communicate with subarachnoid spaces (SAS) directly, whereas for other PVSs, though tangible evidence supported the fact that they must communicate with SAS, a specific anatomical pathway linking the two was to be determined [22]. Importantly, the outer wall of PVS facing the brain parenchyma was lined by astrocyte endfeet expressing Aquaporin4 (AQP4) channels.

In 2012, Iliff and his colleagues firstly named this potential brain waste clearance system including PVSs the “glymphatic system” (GS) (Figure 2) [8]. GS consisted of three main components: CSF influx along periarterial space, exchanges between CSF and ISF in the brain parenchyma, and ISF efflux along perivenous space [8]. Since then, this size-dependent (<20 nm clefts between tight junctions of astrocytes), unidirectionally polarized (influx from periarterial spaces, efflux from perivenous spaces), and spatiotemporally regulated (PVSs share region-specific features and are influenced by circadian, respiratory, etc.) transporting system was recognized as vital for CSF homeostasis, maintaining protein clearance, nutrients distribution, and normal brain signaling [9].

### 2.3. Meningeal Lymphatic Vessel

The earliest records about meningeal lymphatic vessels date back to the 18th century when an Italian anatomist named Paolo Mascagni depicted the existence of dural lymphatic vessels in a wax anatomical model of the human body and organ [23]. However, this viewpoint that lymphatic vessels presented in the dural mater of humans has been overlooked for almost two hundred years [24]. Up until the 20th century, a series of studies that “rediscovered” the presence of mLVs destroyed the silence but were met with questions [25]. In 2015, Aspelund et al. and Louveau et al. identified classic protein markers of lymphatic endothelial cells expressed in meningeal vessels, and thus identified the meningeal lymphatic system of CNS in mice with strong evidence (Figure 3) [6,7].

Absinta et al. firstly showed that the dura mater of human beings contains lymphatic vessels [26]. After the intravenous injection of gadobutrol, they visualized the collection of interstitial gadolinium within mLVs in healthy volunteers. Their results suggested that in the dura, similar to many other organs throughout the body, small intravascular molecules extravasated into the interstitium and then, under a hydrostatic pressure gradient, collected into lymphatic capillaries through a loose lymphatic endothelium. This in vivo finding was further supported by histopathological evidence in humans and marmosets.

Functionally, different anatomical localization of mLVs in the skull of mice showed distinct morphological features and abilities of CSF drainage. Basal mLVs, composed of mature, discontinuously sealed zipper- and button-like junctional patterns of lymphatic endothelial cells (ECs) but lacking smooth muscle cells coverage, were more suited for uptake and outflow CSF than dorsal mLVs with undeveloped morphology [27]. Anatomically, the meningeal lymphatic system of CNS could be described as a drainage network extending from the superior sagittal sinus and the transverse sinuses to the deep cervical lymph nodes (dCLNs) [26,28]. Although mLVs were associated with dural venous sinuses, the precise localization was still not clear, and it is suspected that mLVs were located in dural leaflets, on the inner layer of the dural meter, or in the SAS with the cortical veins [29].

### 2.4. The Interaction between the Glymphatic System and Meningeal Lymphatic Vessels

The relationship between the glymphatic system and the meningeal lymphatic system should be viewed from an integral perspective, namely “the glymphatic-lymphatic system”. Firstly, the dynamic outflow of CSF, which was drained into brain parenchyma along PVSs and was discharged across the dura into mLVs to dCLNs, should be seen as a continuous route. Secondly, the consistent function and contribution of clearance of brain solute and maintaining metabolic balance and neuroimmune homeostasis were indicative of a holistic glymphatic-lymphatic system. Thirdly, previous studies showed that with aging, both the glymphatic system and meningeal lymphatic system become impaired [30,31,32]. The simultaneous dysfunction of the glymphatic system and meningeal lymphatic system implied that even under pathological conditions these two systems contributed together to brain fluid drainage. Fourthly, a previous study showed rapid lymphatic drainage limited glymphatic outflow in conscious rodent mice, supporting the relationship between glymphatic system and meningeal lymphatic system [33]. Hence, it must be emphasized that an overall concept about the connection between the glymphatic system and meningeal lymphatic drainage, and future studies should identify the synergetic roles of these two systems under normal or pathological statuses.

### 2.5. Meningeal and Skull Bone Marrow Immunity

The central nervous system is an immunologically privileged organ, or so a classic theory posited several decades ago. Furthermore, due to the blood-brain barrier (BBB), the interaction between local brain tissue and the peripheral immune system was limited. The discovery of GS and the rediscovery of mLVs shattered this “silence”, as described above. The growing body of evidence indicated rich, diversified immune cells were also populated at the borders of CNS, including dural meninges and cranial bone marrow. It made a stronger voice and opened a novel road to immune surveillance of CNS. Under homeostasis, dural meninges and skull bone marrow naturally hosted a substantial pool of immune cells, such as T cells, macrophages, dendritic cells, mast cells, neutrophils, monocytes, macrophages, and B cells [34,35,36]. Under the conditions of diseases, these immune cells infiltrated brain parenchyma from CNS borders through the direct vascular-like channels and further triggered a series of immune responses in neurons [37]. For example, T cells at the CNS borders modulated behavior and cognition by releasing cytokines in mice [38]. When stroke or aseptic meningitis occurred, neutrophils that reside in skull bone marrow were more likely to rapidly migrate to the adjacent brain tissue than peripheral bone marrow-derived neutrophils [39].

Interestingly, one recent research study showed CSF could permeate into skull bone marrow along perivascular spaces of dural blood vessels [40]. This not only provided a fourth pathway for CSF transport, but also demonstrated a source of information about brain healthy and injury for immune surveillance of CNS borders. From this perspective, CSF was an underestimated messenger that may coordinate neuroinflammation and neuroimmune. It is important to note that the evidence on this is currently insufficient, and future studies should focus more on the role of dural meninges and skull bone marrow in nervous system diseases.

## 3. The Underlying Role of the Glymphatic and Meningeal Lymphatic System in CSVD

### 3.1. Common Vascular Risk Factors

The impairment of glymphatic transport and meningeal lymphatic drainage in the brain has been proven with cerebrovascular risk factors, such as aging, hypertension, diabetes, and lipids metabolism. Since these conditions are strongly linked to CSVD, this also supports the underlying role of glymphatic and meningeal lymphatic dysfunction in the development of CSVD.

#### 3.1.1. Glymphatic-Meningeal Lymphatic Dysfunction in Aging

Aging is deemed as having a key role in regulating the glymphatic and meningeal lymphatic function, and aging-related impairment of the glymphatic–meningeal system has been proven in many studies. In aged mice, advancing age was associated with a dramatic decline in the clearance of intraparenchymal-injected amyloid-beta peptides (A-β), reduction in the vessel wall pulsatility of intracortical arterioles, and widespread loss of perivascular AQP4 polarization [31,32]. During aging, the impairment of glymphatic and meningeal lymphatic functions aggravated cognitive decline and neurodegenerative pathology. Recently, the dysfunction of the glymphatic–meningeal lymphatic system was also observed in older individuals by a novel contrast-enhanced MRI to simultaneously visualize glymphatic and meningeal systems and dCLNs [30]. With an aging population, the incidence of CSVD is continually rising and the association between aging and CSVD is under rising scrutiny. Although aging may not be deemed as a pathology, there is considerable overlap between aging with the process of CSVD. Numerous studies suggested an increased MRI performance of CSVD in the brain of healthy aging humans [41,42,43]. Since the glymphatic-meningeal lymphatic system also declined with aging in the absence of disease, it could be challenging to distinguish the effect of aging from specific processes of CSVD in glymphatic-meningeal lymphatic failure.

#### 3.1.2. Glymphatic-Meningeal Lymphatic Dysfunction in Hypertension

Hypertension, the most robust vascular risk factor for CSVD, causes suppression of the glymphatic function. MRI analysis showed impaired glymphatic clearance of a contrast agent in spontaneously hypertensive rats (SHRs) compared to normotensive rats, indicating that the glymphatic system was impaired during evolving hypertension in early SHR, an effect that worsens in states of chronic hypertension [44,45]. Although the mechanisms of hypertension causing small vascular disease have been widely discussed, glymphatic dysfunction could be regarded as a neglected player in the relationship between hypertension and CSVD.

#### 3.1.3. Glymphatic-Meningeal Lymphatic Dysfunction in Diabetes

Type-2 diabetes mellitus (DM2), a common metabolic disease in middle-aged and older-aged populations, confers an increased risk of CSVD [41,42,46]. MRI analysis using a contrast agent revealed that clearance of the CSF tracer decreased and areas and signal intensity of perivascular arterial influx increased in DM2 rats, compared to non-DM rats [47]. Fluorescent imaging analysis further confirmed the MRI findings. Interestingly, this study displayed glymphatic influx enhancement in DM2; in contrast, all previous studies showed the reduction of the glymphatic influx in CNS diseases. This novel discovery allowed us to speculate that glymphatic pathologies are not only on the side of impairment but also on the side of overstimulation. In addition, cognitive impairment was also correlated to the retention of an MRI contrast agent and fluorescent tracer in DM2 rats, illustrating that a dysfunction of the glymphatic CSF flow results in DM2-induced cognitive deficits [47]. In addition, the activity of glymphatic transport also decreased in patients with DM2 [48].

#### 3.1.4. Glymphatic-Meningeal Lymphatic Dysfunction in Lipids Metabolism

Cholesterol was an essential requirement for the activities of neurons and astrocytes [49]. The metabolic disturbance of blood lipids as well-established vascular factors seemed to have a potential role in small vessel disease, especially high-density lipoprotein [39]. Lipid-modifying therapies have shown benefits in the prevention of ischaemic small vessel disease and leukoaraiosis [50]. The apolipoprotein E(APOE) as the main carrier regulated the transport and metabolism of cholesterol in the periphery and CNS, and APOE polymorphism was strongly related to neurodegeneration and dementia [51]. The CNS relied primarily on production and secretion of APOE from astrocytes and activated microglias [52]. Brain APOE was also derived from CSF or choroid plexus, and the glymphatic system delivered and distributed APOE via the perivascular spaces into neurons [53]. Transport of APOE along periarterial spaces was facilitated by AQP4 at the endfeet of astrocytes [53]. RNA-Sqe analysis showed that induced pluripotent stem cells carrying the APOE4 allele express lower levels of genes related to lymphatic markers, implying APOE4 might play a key role in meningeal lymphosclerosis (e.g., the premature shrinkage of mLVs) and then result in impaired meningeal lymphatic clearance [54]. The finding that selective reduction of astrocytic APOE4 strongly protected against tau-mediated and A-β-accumulated neurodegeneration also indirectly supported this evidence [55,56]. Taking into account these evidences, we proposed a new insight that APOE4-induced cholesterol dysregulation may herald a novel mechanism of CSVD by regulating the dysfunction of the glymphatic-mLVs system.

### 3.2. Mechanisms Linking Glymphatic and Meningeal Lymphatic System to CSVD

#### 3.2.1. Neurovascular Unit Dysfunction and Neurovascular Coupling Dysfunction

The neurovascular unit (NVU) is comprised of neurons, astrocytes, microglia, myocytes, pericytes, ECs, and basement membranes, which describe how these cellular components of the parenchyma and cerebrovascular work together to maintain a brain’s adequate metabolically activity [57]. NVU plays a vitally important role in regulating enough cerebral blood flow (CBF) and neurovascular coupling. As indicated by recent epidemiological, clinical, and preclinical research, the function of NVU cellular components was affected during different stages of CSVD [58]. Notably, the NVU and glymphatic transport have been considered as an interconnected and complementary entity, as they show overlapping physiological functions and closely interact with each other at multiple levels [59]. At the microscopic level, brain activities, vascular pulsation, regional CBF, and generation and clearance of metabolic wastes mediated by NVU are closely matched, which again emphasizes the vital function of glymphatic transport for the integrity of neurovascular coupling and vice versa. At the macroscopic level, arterial pulsation and respiratory and cardiac cycles drive the transport of CSF and ISF along perivascular spaces [45,60,61]. Rhythmic cardiac cycle and respiratory regulate arterial pulsation and maintain vascular continuity from the heart to the brain. Impaired cardiac function contributes to decreased cerebrovascular vessel elasticity and compliance, and further weakened glymphatic flow [45,62]. Hence, the NVU and GS form a structural and functional continuum to maintain the homeostasis of CBF and CSF, and both of them should be deemed important when considering comprehensive viewpoints on the pathophysiology of CSVD.

#### 3.2.2. Blood-Brain Barrier Damage

The blood-brain barrier consists of ECs, pericytes, astrocytic endfeet, and basement membrane, which form a defense against blood-borne pathogens and toxins to parenchyma and strictly controls fluid and nutrient exchange [63,64]. BBB disruption has been confirmed in the pathogenesis of CSVD, which not only caused myelin formation and repair but also caused impaired A-β transport [63,65]. In 2018, the interaction between BBB and GS was reviewed by Verheggen et al. [66]. Firstly, BBB and GS formed a complementary mechanism in the transport and clearance of interstitial solutes and fluid. Specific efflux transporters assisted metabolic wastes such as A-β and Tau to cross the local BBB [67,68]. However, excess protein wastes, which exceeded the transport capacity of BBB, were removed through the glymphatic and meningeal lymphatics vessels to dCLNs [31]. Secondly, the glymphatic dysfunction affected BBB causing cellular dysfunction. The impairment of glymphatic transport caused the loss of AQP4 polarization in astrocytic endfeet and reactive astrogliosis along PVSs [69,70]. The microstructure and the pulsatility of cerebral arterial around the pial sheath and the perivascular astrocytic endfeet sheath were damaged [66]. In addition, the damage of BBB affected the glymphatic function via thickening of the basement membrane and narrowing of the perivascular space. In summary, the glymphatic dysfunction and the BBB disruption created a vicious circle that may exacerbate the burden of small vessel disease.

#### 3.2.3. Neuroinflammation

Neuroinflammation was increasingly deemed as one of the important pathologies of CSVD, with supporting evidence coming mostly from clinical studies and preclinical studies. Microglia, as resident innate immune cells in the brain, generally initiates neuroinflammation and interacts with other cells, especially astrocytes [71,72]. Microglial activation, observed by positron emission tomography imaging of the translocator protein [11C] PK11195, was related to CMBs and WMH in patients with CSVD [73]. Moreover, systemic inflammation markers (e.g., C-reactive protein, homocysteine, and multiple cytokines) and peripheral blood immune cellular phenotype (e.g., CD14+CD16+monocytes) and function (e.g., a shift towards intermediate monocytes and functional reprogramming of peripheral blood mononuclear cell) were associated with the severity and progression of CSVD [58]. Preclinical studies observed enhanced microglial activation, increased oligodendrocyte density, and infiltration of T cells and neutrophils in animal models of CSVD [74,75].

The drainage of immune cells, cytokines, and CNS-derived antigens and antibodies from brain parenchyma to dCLNs along PVSs and mLVs has redefined the CNS as “an immunological privileged organ”. Changes in glymphatic and meningeal lymphatic function are prominent players of neuroinflammation, although several pieces of evidence came from brain tumors and neurodegenerative diseases [76,77,78,79,80,81]. Impaired glymphatic and meningeal lymphatic drainage exacerbate the inflammation burden and microglia response, and promote lymphatic clearance by using vascular endothelial growth factor C (VEGF-C), which could alleviate neuroinflammation as indicated by significantly decreases in IL-1β, INF-γ, and TNF-α and microglia activation [77,79,81,82]. Therapeutic delivery of VEGF-C by mLVs could affect the efficacy of anti-A-β immunotherapy and promote immunosurveillance of brain tumors [76,80]. Although few data have been scant for CSVD, a new therapeutic approach to treat CSVD and vascular cognitive impairment (VCI) has focused on the capacity of VEGF-C to relieve neuroinflammation via enhanced glymphatic and meningeal lymphatic function.

#### 3.2.4. Proteostasis and Protein Aggregation

The gradual imbalance of the various proteostasis machineries, frequently in combination with disorders of protein processing and aggregation through mutated, misfolding, hyperphosphorylated proteins, is increasingly recognized as important pathological mechanisms of the aging brain, especially in neurodegenerative diseases. Impaired proteostasis and protein aggregation might also contribute to CSVD, most prominently cerebral amyloid angiopathy (CAA) and cerebral autosomal-dominant arteriopathy with subcortical infarcts and leukoencephalopathy (CADASIL) [83,84].

Impairment in glymphatic and meningeal lymphatic clearance were key contributors of protein aggregation in brain. In the APP/PS1 model of AD, impaired glymphatic clearance increased A-β deposition, and infusion of A-β into CSF impaired glymphatic drainage in turn, indicating a mutually toxic effect of protein deposition and glymphatic function [32,85]. Decreased meningeal lymphatic outflow was observed in mice injected with alpha-synuclein (α-syn), and blocking CSF outflow via mLVs increased α-syn deposition [86]. More importantly, the prion-like spread pattern of misfolded and aggregated protein was in keeping with the glymphatic flow reflected by neuroimaging [87]. Overall, this evidence strongly supported reduced glymphatic and meningeal lymphatic drainage and might be predicted to increase the risk of protein deposition. The modulation of the balance between protein clearance and aggregation via the glymphatic and meningeal lymphatic system and BBB might present opportunities for AD and CSVD.

#### 3.2.5. Chronic Hypoperfusion and Cerebral Blood Flow Reduction

Chronic cerebral hypoperfusion and cerebral blood flow reduction are common and key mechanisms leading to small vessel disease and VCI [88]. Sustained hypoperfusion in the brain was suggested to be the major cause of WMH and CSVD [89]. In patients with CSVD, hypoperfusion appeared in the WMH and the proximity of the WMH, which performed normally on MRI [90]. Experimental data also suggested hypoperfusion-induced white matter injury [74,91]. Using a model of chronic cerebral hypoperfusion, impaired glymphatic transport was associated with decreased cerebral perfusion, and treatment of digoxin-rescued cerebral perfusion and glymphatic transport [92]. It suggested that glymphatic dysfunction played a vital role in linking hypoperfusion, CSVD, and cognitive impairment.

#### 3.2.6. Hemodynamic Dysfunction

Hemodynamic dysfunction, reflected by declined cerebrovascular reactivity, decreased CBF, changed vascular, and CSF pulsatility, underpinned the progression of CSVD [93,94,95]. On the one hand, the altered hemodynamics of CSVD affected the activity of glymphatic transport. EPVS, as an indicator of impaired fluid drainage in the brain, was associated with reduction in arterial pulsation, which was an important pump for CSF influx into the parenchyma and exchange between CSF-ISF [20]. Preclinical studies suggested the glymphatic dysfunction caused by declined cerebral vascular pulsatility [31,96]. Meanwhile, altered cerebrovascular pulsatility, low cerebrovascular reactivity, and arterial stiffness were associated with EPVS in patients with CSVD [93,97,98]. On the other hand, lower CSF pulsation at the foramen magnum was related to lower cerebrovascular reactivity and worse EPVS, implying reduced PVS flushing may lead to decreased clearance of metabolic waste and PVS enlargement [93]. In conclusion, most studies supported that there is a dynamic interaction between the hemodynamic and glymphatic and meningeal lymphatic systems in CSVD; therefore, future studies should add direct evidence to support this hypothesis.

#### 3.2.7. Venous Collagenosis

Venous collagenosis (VC), characterized by the thickening of the vascular walls and narrowing of the lumen as a result of collagen deposition, was a pathology of CSVD [99]. The disruption or decreased visibility of deep medullary veins (DMVs) on susceptibility-weighted imaging has been regarded as the neuroimaging marker of VC in patients with CSVD [100]. A recent study found that DMVs were associated with EPVS in basal ganglia, suggesting that the disruption of DMVs was involved in glymphatic fluid stasis in the perivascular spaces of basal ganglia [101]. No association between DMVs and EPVS in centrum semiovale was found; the potential explanation was that cortical and superficial veins and venules drained CSF/ISF in centrum semiovale where the glymphatic function was still functioning normally or in a compensatory phase [101]. As perivenous space plays a key role in the glymphatic efflux, whether the deposition of collagenosis in venous affects the glymphatic system needs more direct evidence in the future.

### 3.3. Subtype of Cerebral Small Vessel Disease

#### 3.3.1. Amyloid-Related Cerebral Small Vessel Disease

Amyloid-related CSVD, also namely cerebral amyloid angiopathy, is characterized by the progressive aggregation and deposition of A-β within the basement membranes of small leptomeningeal and cortical arteries [102]. CAA is thought to be caused by the failure of intramural periarterial drainage (IPAD) and the perivascular CSF influx [102]. Glymphatic and meningeal lymphatic impairment increased the risk of presenting with CAA as they decrease A-β clearance. A pre-clinical study showed IPAD and CSF transport among the glymphatic and meningeal lymphatic system were impaired in the APP/PS1 mice and further aggravated in old age, implying that the severely impaired CSF tracer influx and efflux patterns exist in CAA [103]. CAA was not only a consequence of decreased glymphatic-meningeal lymphatic clearance, but also a contributor in this process, causing impairment of arterial function and a change in perivascular structure as well as increasing the severity of AD pathology. A recent study by Chen et al. found CSF transport to dCLNs occurred along the carotid arteries and was time-delayed in CAA, indicating that upstream connections to the meningeal lymphatic drainage were damaged [104].

#### 3.3.2. Arteriolosclerosis Cerebral Small Vessel Disease

Arteriolosclerosis cerebral small vessel disease (aCSVD), also called age-related and vascular-risk-factor-related CSVD, is the most major subtype of small vessel disease, mainly characterized by loss of smooth muscle cells from the tunica media, accumulation of fibro-hyaline material, narrowing of the lumen, and thickening of the blood vessel wall [99]. Bilateral common carotid stenosis (BCAS), by induced chronic cerebral hypoperfusion and white matter lesions, was a typical mouse model of aCSVD and VCI [91]. BCAS mice showed decreased pulsation of cortical vessels and glymphatic dysfunction by impaired CSF fluorescent tracer influx three months after surgery [105]. A clinical study investigated glymphatic function by non-invasive DTI-ALPS in patients with CSVD [12]. Neuroimaging changes of CSVD, including WMH, lacunas, CMBs, and EPVS were strongly related to DTI-ALPS, suggesting that glymphatic drainage impairment may lead to the development of aCSVD [12]. Another clinical study also found these neuroimaging features above were associated with a decreased DTI-ALPS index [106]. In the future, larger prospective studies should investigate the influence of the glymphatic and meningeal lymphatic system in the progression of aCSVD.

#### 3.3.3. Microinfarcts

Microinfarcts usually occur with aging as consequence of diffuse embolic events and are correlated with vascular dementia. A rodent study showed that multiple microinfarcts by intraarterial injection of cholesterol crystals resulted in a global impairment of CSF influx along the glymphatic pathway [107]. Although the glymphatic dysfunction was transient, with recovery within 2 weeks of injury, CSF tracers still accumulated within local brain tissue associated with multiple microinfarcts, illustrating a more subtle and focal impairment of glymphatic pathway function that may continue long after the recovery of global glymphatic CSF flow [107]. Additionally, the impact of microinfarcts on glymphatic system dysfunction was aggravated in 12-month-old mice compared to 2–3- month-old mice, suggesting that glymphatic function was more vulnerable in the aging brain with microinfarcts than the young brain [107]. These findings implied that microinfarcts may trap solutes and proteins within parenchyma, increasing the risk of the formation of Aβ plagues and the development of vascular dementia.

### 3.4. Cognitive Impairment and Dementia: A Common Final Pathway

Impaired cognition in hereditary CSVD, such as CADASIL and cerebral autosomal recessive arteriopathy with subcortical infarcts and leukoencephalopathy (CARASIL), has demonstrated findings to be qualitatively similar to those with performance in sporadic CSVD but with more severe performance in the genetic types. It indicated that CSVD of different origins had a common final pathway toward cognitive impairment and dementia. In fact, a number of researches confirmed that CSVD co-occurred with AD or PD, and signs of A-β, tau, and α-syn pathologies were also found in CSVD, supporting the communication between CSVD and neurodegeneration. Kim et al. and Cai et al. have presented detailed summaries of the epidemiological and clinical-pathological data on the association between CSVD and AD, respectively [108,109]. Similar pathophysiological mechanisms, including neuroinflammation, mitochondrial disruption, oxidative stress, and metabolic disorders, also indicated the complex interaction between AD and CSVD. The glymphatic and meningeal lymphatic function decreased with aging, and the glymphatic and meningeal lymphatic failures were antecedents to the onset of dementia in neurodegenerative diseases such as AD and PD [19,31,32,76,85,110,111]. For these reasons, we turn our attention to the function of glymphatic-meningeal lymphatic system, since the glymphatic-mLVs transport system may yield a breakthrough in understanding the interconnection between CSVD and neurodegenerative disease (Figure 4).

In addition, sleep played a prominent role in resistance to neurodegeneration and small vessel disease, and the disturbed sleep-wake cycle increased the risk of both AD and CSVD. Sleep deprivation contributed to the accumulation of metabolic wastes and toxins in the brain and led to dramatic glymphatic and meningeal lymphatic impairment [112,113]. Thus, sleep regulation provided a promising therapeutic opportunity for CSVD and AD. A better understanding of the link between sleep disturbances and reducing of CSF and protein clearance associated with cognitive decline in AD and CSVD may shed light on their association. The methods of improving sleep quality, especially in the non-rapid eye movement sleep period, can open a new era in strategies of early therapy of CSVD, including the potential risk to develop into cognitive disorders and neurodegenerative diseases in the future.

## 4. Conclusions

In the present review, we have pointed out four main aspects linking glymphatic and meningeal lymphatic systems and CSVD, including vascular risk factors, potential pathological mechanisms, clinical subtypes, and dementia. Based on our findings, we propose above changes in the function of GS and mLVs, which could be a prominent feature of CSVD and could be suggested as an early pathological marker of this condition. The precise pathway in glymphatic and meningeal lymphatic failure resulting in CSVD remains unclear, but growing evidence has indicated that the glymphatic-meningeal lymphatic system may play a contributing role in the development of CSVD. To provide an overall insight into the pathological process between glymphatic and meningeal lymphatic dysfunction with CSVD, we presented seven potential pathological mechanisms separately. In fact, these mechanisms are not self-containing; there is a dynamic interaction among these mechanisms. Although there is a complex interaction between glymphatic and meningeal lymphatic systems and several pathologies of CSVD, the most current findings have focused on a single pathological mechanism. It is for this reason that the exact sequence and relative significance of initial events of CSVD are still elusive in humans. Ideally, longitudinal observational studies are needed to assess temporal changes in glymphatic and meningeal lymphatic function during aging and throughout the progression of CSVD. As the underlying interaction is becoming more explicit, several evidence gaps are in urgent need to be filled in. Moreover, most evidence for these pathological mechanisms involved in glymphatic and meningeal lymphatic dysfunction in CSVD comes from rodent studies, which are considered as low level evidence, since it is unknown how findings in animal models associate with findings in patients with CSVD. Hence, both the glymphatic system and meningeal lymphatic system may be compromised in the process and development of CSVD, and both systems represent novel therapeutic targets for treatment of CSVD-related cognitive decline and dementia.

## Figures and Tables

**Figure 1 biomolecules-12-00748-f001:**
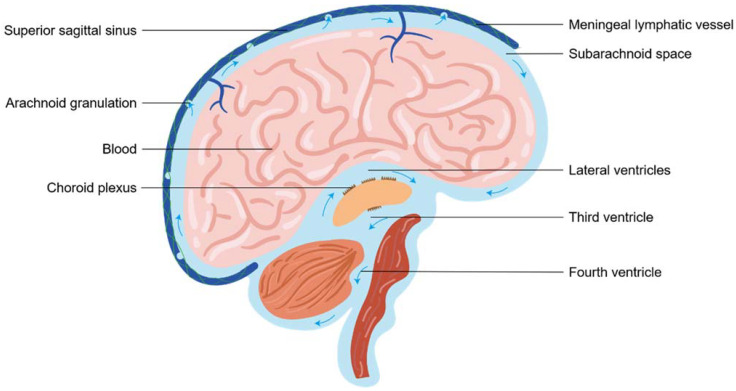
A brief overview of cerebrospinal fluid circulation. Traditionally, Cerebrospinal fluid (CSF) produced in the choroid plexus flows from later ventricles, the third ventricle and fourth ventricle to the subarachnoid space (SAS) of the brain. In the SAS, CSF was reabsorbed through the arachnoid granulations into the venous sinuses for efflux. In addition, the perineural sheaths surrounding cranial nerves through the cribriform plate also could drain CSF. Recent advances suggested meningeal lymphatic vessels drained CSF to deep cervical lymph nodes. The contribution and importance of these CSF outflow pathways are debated.

**Figure 2 biomolecules-12-00748-f002:**
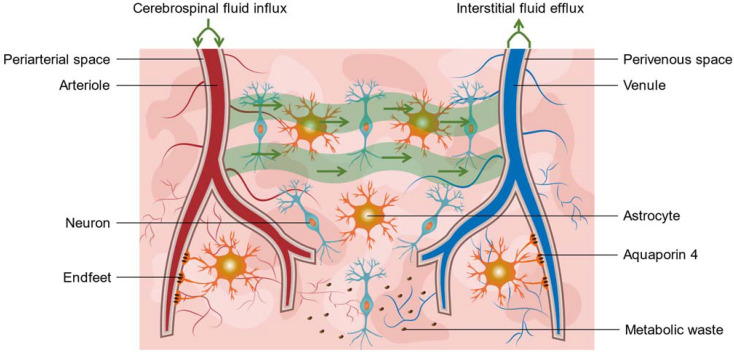
A brief overview of the glymphatic system. The glymphatic system consisted of three main components: cerebrospinal fluid (CSF) influx along periarterial spaces, exchange between CSF and interstitial fluid (ISF) in the brain parenchyma, and ISF efflux along perivenous spaces. Aquaporin4 (AQP4) located in astrocyte endfeet toward perivascular spaces were essential to maintain the normal function of the glymphatic transport, such as metabolic waste clearance. The loss of AQP4 polarization resulted in glymphatic failure in aging.

**Figure 3 biomolecules-12-00748-f003:**
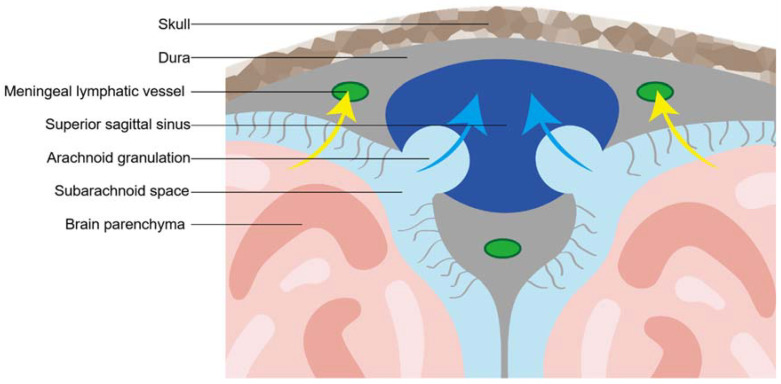
A brief overview of meningeal lymphatic vessels. Meningeal lymphatic vessels (mLVs) could be described as a lymphatic network in the central nervous system that expresses classic protein markers of lymphatic endothelial cells, such as lymphatic vessel endothelial hyaluronan receptor 1 (LYVE1), vascular endothelial growth factor receptor 3 (VEGFR3), prospero homeobox protein 1 (Prox1), and chemokine (C-C motif) ligand 21 (CCL21). Developmentally, the maturity of mLVS depend on vascular endothelial growth factor C (VEGF-C). Anatomically, mLVs extend from the superior sagittal sinus and the transverse sinuses to into the deep cervical lymph nodes (dCLNs). Functionally, mLVs play a significant role in draining of CSF, metabolic waste, immune cells, or other substances.

**Figure 4 biomolecules-12-00748-f004:**
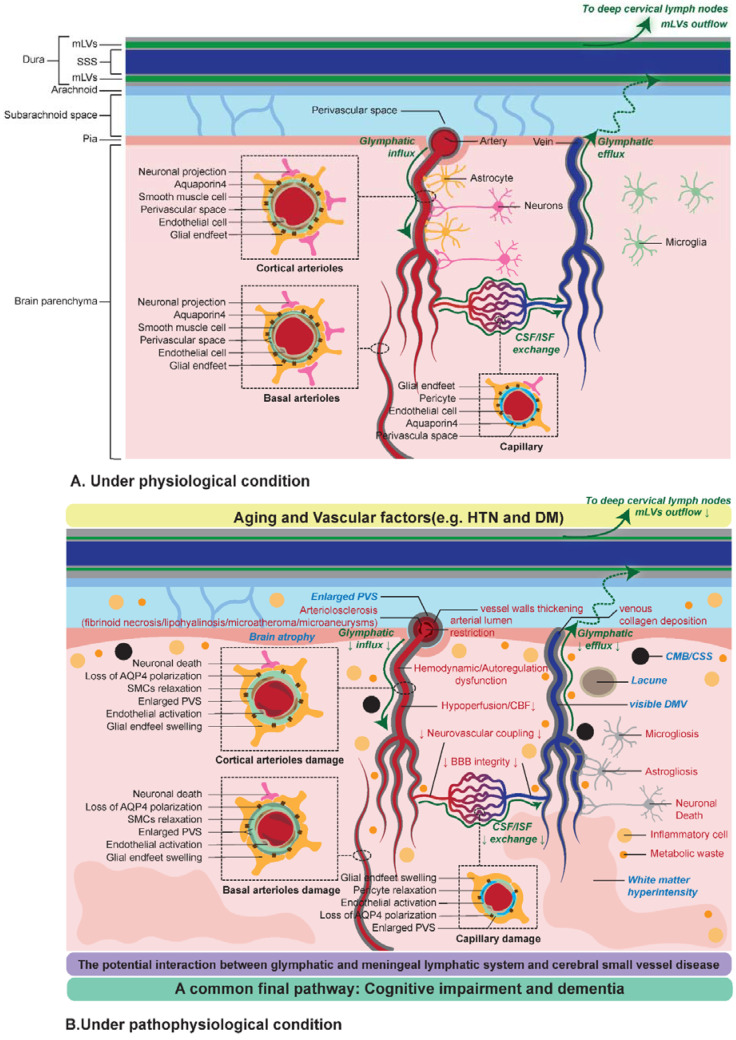
The possible hypothesis of potential interaction between CSVD and glymphatic and meningeal lymphatic system. In sporadic CSVD, normal aging may be an initial event in CSVD pathobiology. Under the conditions of normal aging, vascular risk factors (i.e., hypertension and diabetes), and genetic factors cause a series of structural and functional changes in the brain and cerebral small vessels occur. There are four main aspects of pathological and pathophysiological mechanisms involved in CSVD: cerebral small vascular cellular damage, abnormal structure of cerebral small vessels, abnormal regulation of cerebral small vessels, and cerebral homeostasis imbalance. Although we proposed a structured insight of the pathological and pathophysiological mechanisms of CSVD, in fact, these processes are not independent, and there is a continuous dynamic interaction among these mechanisms. The pathophysiological processes of CSVD were related with glymphatic and meningeal lymphatic failure. Neuroimaging features on MRI are direct evidence of CSVD, including cortical superficial siderosis, cerebral microbleeds, cortical microinfarction, lacunas, white matter hyperintensity, brain atrophy, enlarged perivascular space (EPVS), and deep medullary veins. PVSs are important composite parts of the glymphatic system, whereas EPVSs visible on MRI indicate impaired glymphatic and meningeal lymphatic transport. A vicious circle was created between the glymphatic and meningeal lymphatic dysfunction and CSVD that may finally result in cognitive impairment and dementia.

## Data Availability

Not applicable.

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
