# Peer review of "The Underlying Role of the Glymphatic System and Meningeal Lymphatic Vessels in Cerebral Small Vessel Disease"

_biomolecules, 2022, doi:10.3390/biom12060748_

Round 1
Reviewer 1 Report
I have doubts about the methodology of writing the work. what criteria did the authors use when selecting references?
how many articles have been analyzed? What databases were used for literature screening? the authors write that they provide the latest reports, the newest, i.e. from what period?
Author Response
Dear reviewers:
We would like to thank you for your careful reading, helpful comments, and constructive suggestions, which has significantly improved the presentation of our manuscript.
We have carefully considered all comments from the reviewers and revised our manuscript accordingly. The manuscript has also been double-checked, and the typos and grammar errors we found have been corrected. A native speaker has corrected this manuscript. And we have revised the format of references.
In our revisions, we paid specific attention to (1) add a graphical abstract, (2) modify introduction, (3) add Section 2.5. Meningeal and skull bone marrow immunity, (4) add Section 3.1.4. Glymphatic-meningeal lymphatic dysfunction in lipids metabolism, and (5) revise Section 3.4. cognitive impairment and dementia. This is a brief summary of our revision. In the following section, we summarize our detailed responses to each comment from the reviewers.
We believe that our responses have well addressed all concerns from the reviewers. We sincerely hope our revised manuscript can be accepted for publication in Biomolecules.
Thank you for pointing out this problem in manuscript again.
Best wishes,
Yilong Wang and Yu Tian
Beijing Tiantan Hospital of Capital Medical University, Beijing, China
#Reviewer 1
Comment: I have doubts about the methodology of writing the work. what criteria did the authors use when selecting references? how many articles have been analyzed? What databases were used for literature screening? the authors write that they provide the latest reports, the newest, i.e. from what period?
Response: Thank you for your comment. In writing this review, we started by listing the entire framework based on our understanding of cerebral small vessel disease. Then, we read a mass of literatures and eventually screened most of the studies and reviews that we considered to be of important. We conducted a literature search of PUBMED database and EMBASE database. We used headings related to Cerebral Small Vessel Disease, Glymphatic System, Meningeal Lymphatic System, and their related alternatives from 2012 up to May 2022. In current revision, the newest reference was published on 2 May 2022.
The point I need to explain is that our article is a knowledge review, not a systematic or rapid review. We acknowledge that this is a shortcoming of this review because of the lack of a systematic literature search and qualification evaluation. Unfortunately, there are not much studies about the relationship between cerebral small vessel disease and the glymphatic-meningeal lymphatic system. However, efforts have been made to include more relevant references. There finally were 113 references in this revised manuscript.
Reviewer 2 Report
In principle, this is a very interesting review written for a broad readership, in particular for scientists dealing with dementia or neuroscience in general. The article explains the basic mechanisms and illustrates them nicely in high-quality pictures making it easy to follow the main mechanisms. However, it should be pointed out that the scientific depth of this review is limited and written more for people who want to get a quick overview of the topic.
The review has a clear focus and structure, and the language is readable but I recommend that a native speaker goes through the manuscript and smooth some phrases. (But this is only a minor point).
The cited literature is mostly comprehensive and up to date. Opinion and facts are mostly clearly separated and the topic is resented in a balanced way.
In summary, I have only minor suggestions, which might help to improve the quality of this review:
- I have not found a graphical abstract. This is a pity. According to my experience, a graphical abstract helps a lot that the article is recognized and therefore more cited. In the authors' own interest, I recommend making an overview figure summarizing the underlying mechanisms for a graphical abstract.
- The introduction is very short. I recommend summarizing here some more clinically relevant data and a more detailed overview of the disease.
- To my knowledge, there are some papers published dealing with the impact of lipids. This - in my opinion very important aspect – is completely neglected. I would appreciate if the authors would add a section dealing with this topic. Just to give an example; an important article is recently published in brain: Brain.2020 Feb 1;143(2):597-610. doi: 10.1093/brain/awz413. (But there are several more interesting lipids additional to cholesterol)
- Section 3.4.: I absolutely agree with the authors that there are common mechanisms shared by e.g. AD and CSVD. Unfortunately, none of them are really explained or listed and the section stays very superficial. Please explain the shared mechanisms in a more detailed way.
- As mentioned above the manuscript could benefit from being corrected by a native speaker.
All the changes can be done in a reasonable time without a great effort. I, therefore, recommend accepting this article after minor revisions.
Author Response
Dear editors and reviewers:
We would like to thank you for your careful reading, helpful comments, and constructive suggestions, which has significantly improved the presentation of our manuscript.
We have carefully considered all comments from the reviewers and revised our manuscript accordingly. The manuscript has also been double-checked, and the typos and grammar errors we found have been corrected. A native speaker has corrected this manuscript. And we have revised the format of references.
In our revisions, we paid specific attention to (1) add a graphical abstract, (2) modify introduction, (3) add Section 2.5. Meningeal and skull bone marrow immunity, (4) add Section 3.1.4. Glymphatic-meningeal lymphatic dysfunction in lipids metabolism, and (5) revise Section 3.4. cognitive impairment and dementia. This is a brief summary of our revision. In the following section, we summarize our detailed responses to each comment from the reviewers.
We believe that our responses have well addressed all concerns from the reviewers. We sincerely hope our revised manuscript can be accepted for publication in Biomolecules.
Thank you for pointing out this problem in manuscript again.
Best wishes,
Yilong Wang and Yu Tian
Beijing Tiantan Hospital of Capital Medical University, Beijing, China
#Reviewer 2
Comment 1: I have not found a graphical abstract. This is a pity. According to my experience, a graphical abstract helps a lot that the article is recognized and therefore more cited. In the authors' own interest, I recommend making an overview figure summarizing the underlying mechanisms for a graphical abstract.
Response: We are very sorry for our negligence of the graphical abstract. The graphical abstract should summarize the contents of the article in a concise. In general, a graphical abstract is optional. As graphical abstract is not mandatory, we missed this important part at the beginning submission. Now, we have provided a simple but highlighted graphical abstract which consist of a short collection of bullet points that convey the core findings of this article.
Comment 2: The introduction is very short. I recommend summarizing here some more clinically relevant data and a more detailed overview of the disease.
Response: Thank you for your constructive. In view of the detailed explanation in the later part of the article, in the introductory section we only briefly described cerebral small vessel disease and the glymphatic and meningeal lymphatic vessels. Not much explanation has been given. Now, as an initiate, we have introduced two convincing evidences about the relationship between glymphatic-meningeal lymphatic transport and small vessel disease. One preclinical study revealed that aging spontaneously hypertensive stroke prone rats, a most frequently used model for simulating the pathology of CSVD in human, showed significantly reduced cerebrospinal fluid transport and moderately decreased glymphatic function compared with Wistar-Kyoto rats[11]. Another clinical study illustrated the relationship between glymphatic function reflected by diffusion tensor im-age analysis along the perivascular space and neuroimaging markers, including WMH, lacunas, CMBs, and EPVS in basal ganglia, providing more convincing evidence about the glymphatic-meningeal lymphatic system in small vessel disease[12]. In addition, in the last paragraph of the introductory section, we have re-described structure of this review in order to provide a better reading experience for the readers.
Comment 3: To my knowledge, there are some papers published dealing with the impact of lipids. This - in my opinion very important aspect – is completely neglected. I would appreciate if the authors would add a section dealing with this topic. Just to give an example; an important article is recently published in brain: Brain.2020 Feb 1;143(2):597-610. doi: 10.1093/brain/awz413. (But there are several more interesting lipids additional to cholesterol).
Response: We gratefully appreciate for your valuable suggestion. Dyslipidemia is a key vascular risk factor for ischemic small vessel disease; statins are beneficial for patients with cerebrovascular disease. We previously overlooked this because there are no direct studies of hyperlipidaemia and g lymphatic and meningeal lymphatic system dysfunction.
Previous studies showed the E4 variant of apolipoprotein E (APOE4) is the main susceptibility gene for Alzheimer’s disease, and altering APOE levels in the brain can influence the development of neurodegenerative pathologies, such as A-beta and tau accumulation. We add the Section 3.1.4. Glymphatic-meningeal lymphatic dysfunction in lipids metabolism. We focus on the potential impact of APOE on the clearance of the glymphatic-meningeal lymphatic system and the effect of this impact in the progression of CSVD. Interestingly, RNA-Seq analysis illustrated induced pluripotent stem cells carrying the APOE4 allele express lower levels of genes associated with lymphatic markers, and (b) genes for which well-characterized missense mutations have been linked to peripheral lymphedema[54]. It implied that APOE4 could play a novel role in the premature shrinkage of meningeal lymphatic vessels (meningeal lymphosclerosis), leading to abnormal meningeal lymphatic functions (meningeal lymphedema), and, in turn, reduction in the clearance of amyloid-beta and other macromolecules and inflammatory mediators, as well as immune cells, from the brain, exacerbation of CSVD manifestations, and progression of the disease. So, we have provided a brief review of the relevant literatures in this section.
Comment 4: Section 3.4.: I absolutely agree with the authors that there are common mechanisms shared by AD and CSVD. Unfortunately, none of them are really explained or listed and the section stays very superficial. Please explain the shared mechanisms in a more detailed way.
Response: We have made correction according to the reviewer’s comments. We have restructured Section 3.4 in the revised manuscript. One point that needs to be explained is that in this part we have mainly highlighted the role of vascular cognitive dysfunction and dementia in the outcome of cerebral small vessel disease. The mechanisms involved are described in more detail in Section 3.2. Mechanisms linking glymphatic and meningeal lymphatic system to CSVD, especially in Section 3.2.4. Proteostasis and protein aggregation. As mentioned in the manuscript, Kim et al. and Cai et al. have presented detailed summaries of the epidemiological and clinical-pathological data on the association between CSVD and AD, respectively[108,109].
In addition, sleep disorder was a prominent contributor of the impaired glymphatic-lymphatic vessel function and finally resulted in dementia. Patients with CSVD or Alzheimer's disease showed sleep disturbances, especially in non-rapid eye movement sleep period. The improvement of sleep quality is a novel treatment strategy of dementia in the early stage. We also emphasized this point in revised manuscript.
Comment 5: As mentioned above the manuscript could benefit from being corrected by a native speaker.
Response: Thank you for underlining this deficiency. We are very sorry for the mistakes in this manuscript and inconvenience they caused in your reading. The manuscript has been thoroughly revised and rewritten by a native English speaker. The typos and grammar errors we double-checked have been corrected.
Other response: We note that a recent article has shown that cerebrospinal fluid CSF could permeate into the skull bone marrow along perivascular spaces of dural blood vessels [40]. It provided a new fourth pathway for CSF transport. It is more significant in suggesting an immune surveillance role at the border of the central nervous system, including dural meninges and skull bone marrow. Therefore, we have reviewed the relevant researches[34-40] and offer some of our own views in Section 2.5. Meningeal and skull bone marrow immunity.